https://doi.org/10.1038/s41467-018-07940-1　　**OPEN**

# Contribution of individual olfactory receptors to odor-induced attractive or aversive behavior in mice

Nao Horio[1], Ken Murata[1,2], Keiichi Yoshikawa[1], Yoshihiro Yoshihara[2,3] & Kazushige Touhara[1,2,4]

Odorants are recognized by multiple olfactory receptors (ORs) and induce innate behaviors like attraction or aversion via olfactory system in mice. However, a role of an individual OR is unclear. Muscone is recognized by a few ORs including MOR215–1 and MOR214–3, and attracts male mice. Odor preference tests using MOR215–1 knockout mice revealed that MOR215–1 and other OR(s), possibly including MOR214–3, are involved in the attraction. (Z)-5-tetradecen-1-ol (Z5–14:OH) activates ~3 ORs, including Olfr288, and evokes attraction at low levels but aversion at higher levels. Olfr288 knockout mice show no attraction but aversion, suggesting Olfr288 is involved in preference for Z5–14:OH, whereas activation of other low-affinity Z5–14:OH receptors evokes aversion. Each OR appears to send a signal to a neural circuit that possesses distinct valence, leading to a certain behavior. The final output behavior with multiple ORs stimulation is determined by summation (addition or competition) of valences coded by activated ORs.

[1] Department of Applied Biological Chemistry, Graduate School of Agricultural and Life Sciences, The University of Tokyo, 1-1-1, Yayoi, Bunkyo-ku, Tokyo 113-8657, Japan. [2] ERATO Touhara Chemosensory Signal Project, JST, The University of Tokyo, Tokyo 113-8657, Japan. [3] Laboratory for Systems Molecular Ethology, RIKEN Center for Brain Science, Saitama 351-0198, Japan. [4] International Research Center for Neurointelligence (WPI-IRCN), The University of Tokyo Institutes for Advanced Study, Tokyo 113-0033, Japan. Correspondence and requests for materials should be addressed to K.T. (email: ktouhara@mail.ecc.u-tokyo.ac.jp)

Animals use up to a thousand odorant receptors (ORs), expressed in the olfactory epithelium, to detect a wide range of odorant molecules in the external world[1]. In general, each odorant is recognized by dozens of ORs in a combinatorial fashion[1,2]. For example, measurements of the odorant responses of dissociated olfactory sensory neurons revealed that octanal is recognized by more than 30 ORs in rat[3]. Eugenol (EG) is recognized by about 45 ORs, as demonstrated by imaging and c-Fos mapping of the olfactory bulb[4–7]. By contrast, odorants such as androstenone and muscone are recognized by a relatively small number of ORs[5,8,9]. In any case, typical ORs are functionally redundant, although each OR has a distinct ligand spectrum, suggesting that this redundancy contributes to the discriminatory power of the olfactory system.

Because a defective mutation in one OR does not affect the ability of other ORs to detect an odorant, this functional redundancy helps to avoid complete anosmia to specific odorants. However, in humans, genetic variations in single ORs can affect sensitivity to, or the intensity of perception of, their cognate odorants[8,10–14]. Similar observations have also been reported in mice. Genetic deletion of MOR215–1, the most sensitive receptor of muscone, results in reduced sensitivity to that compound[15]. These results suggest that the effect on intensity or sensitivity is most obvious when the targeted OR has the highest affinity for a given odorant among the ORs that recognize it.

Odorants are recognized by multiple ORs in olfactory sensory neurons in the olfactory epithelium. Then the odor information is sent to the olfactory bulb (OB) and various higher brain areas, leading to behavioral outputs. In mice, urinary odorants such as (methyl-thio)methanethiol (MTMT) and (Z)-5-tetradecen-1-ol (Z5–14:OH) attract the opposite sex[9,16]. By contrast, predator odorants such as trimethyl-thiazoline (TMT), secreted from the anal gland of foxes, induce aversive behavior[17–19]. In addition to these semiochemicals utilized for intra- or inter-species communications, many general odorants cause attraction or avoidance behavior in mice[20], suggesting that odorants tend to possess positive or negative valence. Using various genetic tools, the brain areas that mediate attractive or aversive signals have gradually been revealed[21].

The relationship between a single olfactory receptor and a behavioral output has been investigated for a trace amine-associated receptor (TAAR), which belongs to another OR family in the olfactory epithelium. Deletion of TAAR4 eliminates the aversion that mice exhibit toward low concentrations of volatile amines and the odor of predator urine[22], and amine sensitivity is set solely by the most sensitive TAAR[23]. Recent work revealed the relationship between a single OR and a behavioral output. Specifically, activation of Olfr1019, one of the receptors for TMT, induces immobility in mice. In Olfr1019 knockout mice, this immobility response is reduced, but not entirely abolished, due to the presence of other TMT-responsive glomeruli[24]. In general, because one odorant activates multiple ORs, the encoding of odor-associated valence at the level of ORs is complex.

In this study, we investigated how individual ORs are involved in odor-induced attractive/aversive behavior in mice. The fact that odorants are recognized by multiple ORs creates challenges for study at the individual OR level, largely because it would be impractical to simultaneously knockout dozens of ORs. Therefore, we selected two odorants that each activates a small number of ORs in mice. Muscone activates a few ORs including MOR215–1 and MOR214–3[5]. Among them, MOR215–1 is the most sensitive: MOR215–1 knockout mice exhibited a dramatic reduction in sensitivity to muscone[15]. Another odorant, Z5–14:OH, activates ~3 ORs, including Olfr288, and attracts female mice[9]. By conducting a two-choice odor-preference test and an odor investigation assay in mice lacking MOR215–1 or Olfr288, we revealed a link between single ORs and odor-induced behaviors.

## Results

### Involvement of MOR215–1 and MOR214–3 in attraction to muscone in male mice.
We previously reported that muscone activates a few ORs in mice, including MOR215–1 and MOR214–3, of which the former MOR215–1 is more sensitive[5,15]. Both wild type (WT) and MOR215–1 knockout (KO) mice could detect 450 μg muscone. MOR215–1 KO mice could not detect 45 ng muscone, whereas WT males could, suggesting that mice detect the higher amount (450 μg) of muscone via both MOR215–1 and MOR214–3, but the lower amount (45 ng) muscone via MOR215–1 alone[15]. We then performed a two-choice odor-preference test in which mice were presented with a choice between two samples, and the amount of time spent investigating each sample was measured[9] (Fig. 1a). WT female mice did not exhibit any interest in muscone (Fig. 1b), whereas WT male mice nose-poked a vent from a room containing 450 μg muscone significantly longer than control air (Fig. 1c, upper left panel). The basal activities are very different from mouse to mouse, however when we focus on each mouse data, the investigation time of muscone was longer than PG in almost all WT male mice. MOR215–1 KO mice exhibited the same attractive behavior as WT mice. KO males investigated 450 μg muscone longer than control air (Fig. 1c, upper right panel). At 45 ng muscone, an amount that activates MOR215–1 but not MOR214–3 (Fig. 1d, lower panel), the preference was still observed in WT mice (Fig. 1d, upper panel), suggesting that activation of only MOR215–1 evoked the attractive behavior. By contrast, MOR215–1 KO males no longer exhibited a muscone preference, because they are unable to detect 45 ng muscone (Fig. 1d, right upper panel). Together, these results suggest that MOR215–1 and the other receptor(s), possibly including MOR214–3, are involved in attraction to muscone (Fig. 1c, d, lower panels).

### Olfr288 is the most sensitive OR for Z5–14:OH among ~3 ORs in mice.
Z5–14:OH is secreted from the preputial gland in male mice, and activates ~3 ORs in female mice, including Olfr288, leading to attraction[9]. To examine the involvement of Olfr288 the attractive behavior, we generated Olfr288-KO mice. Olfr288-KO mice did not express Olfr288 in the main olfactory epithelium (Supplementary Fig. 1). The KO mice were stimulated with 1 mg and 0.1 ng Z5–14:OH, and the activated glomeruli in the olfactory bulb were counted. Consistent with a previous report[9], 1 mg Z5–14:OH activated four to seven glomeruli (corresponding to ~3 ORs, including Olfr288) and 0.1 ng Z5–14:OH activated one to three glomeruli (corresponding to ~1 ORs, likely including Olfr288) in WT mice (Fig. 2a, Supplementary Fig. 2). In Olfr288-KO mice, 1 mg Z5–14:OH activated three or four glomeruli (corresponding to ~2 ORs), and 0.1 ng Z5–14:OH activated almost zero glomerulus, which is reasonable because the KO mice lack one of the Z5–14:OH receptors, Olfr288 (Fig. 2a, Supplementary Fig. 2). As a control, no glomerulus activation was observed without odorant stimulation. The location of activated glomeruli were similar in different OBs/animals, suggesting that the same set of ORs are likely activated. (Supplementary Fig. 3).

To examine the sensitivity of KO mice to Z5–14:OH, we performed the odor-finding test as described above for the MOR215–1 KO mice. Individual mice were exposed for 5 min to the tip of a glass capillary with various amount of Z5–14:OH during the dark period. In WT female mice, the threshold amount of Z5–14:OH was ~3 pg under this experimental condition: no mouse could detect 1 pg, but more than a half of the tested mice

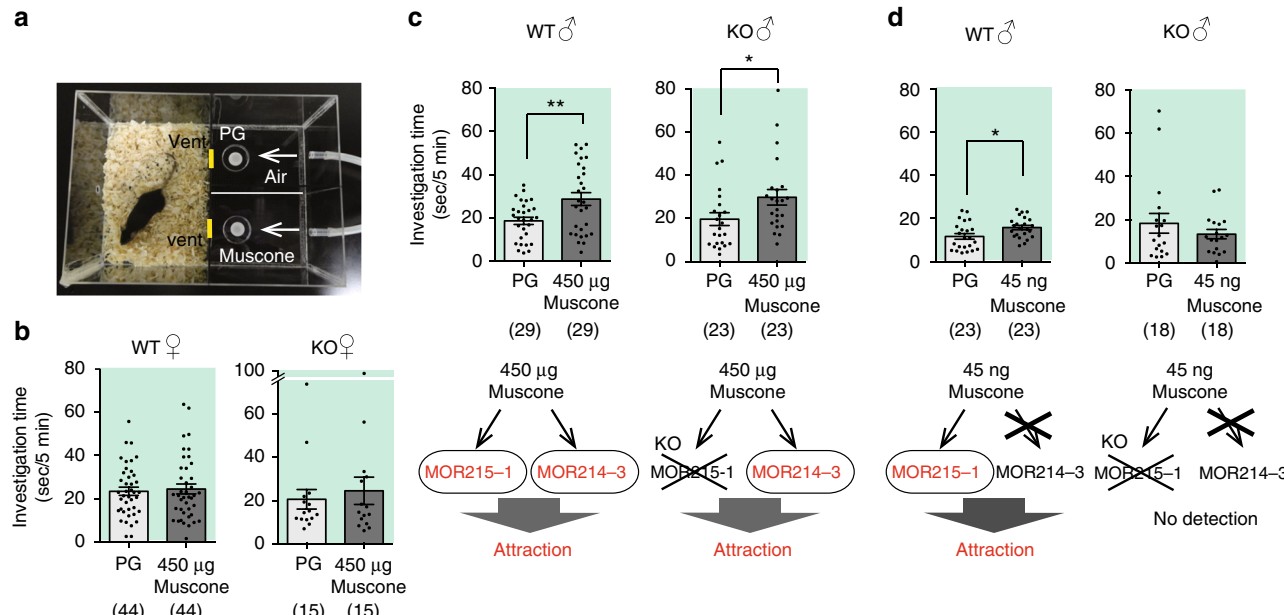

**Fig. 1** Attraction to muscone via MOR215–1 and MOR214–3 in male mice. **a** Setup for the two-choice preference test. An individual test mouse is introduced into the room, and can poke its nose into either of two holes, from which muscone-containing air or control air (PG; propylene glycol) is constantly supplied. The total time spent the mouse spent nose-poking each vent hole was measured. **b** Investigation time for muscone (450 μg) in wild type or MOR215–1 knockout female mice: PG (propylene glycol) vs. 450 μg muscone dissolved in PG in C57BL/6 adult female mice (WT) ($N = 44$) and MOR215–1 knockout adult female mice (KO) ($N = 15$). **c** Investigation time for muscone (450 μg) in wild type or MOR215–1 knockout male mice. (top) PG vs. 450 μg muscone in PG in C57BL/6 adult male mice (WT) ($N = 29$) and MOR215-1-KO adult male mice (KO) ($N = 23$). (bottom) A possible model showing the involvement of two muscone ORs in attraction to 450 μg muscone. **d** Investigation time for muscone (45 ng) in wild type or MOR215–1 knockout male mice. (top) PG and MOR215-1-KO adult male mice (KO) ($N = 18$). (bottom) A possible model showing how 45 ng muscone activates only MOR215–1, leading to attraction. Bars indicate the mean time that a mouse spent investigating each test sample within a 3-min period, ±S.E.M. Asterisks indicate significant differences between two samples (paired Student's $t$-test, *$P < 0.05$, **$P < 0.01$)

could find 3 pg (Fig. 2b). Almost all WT female mice could detect 10 pg Z5–14:OH, whereas almost none of the Olfr288-KO female mice could do so (Fig. 2b). By contrast, the detection threshold for eugenol, a neutral odorant, was similar between WT and Olfr288-KO female mice (Fig. 2c). These results indicate that the threshold amount for Olfr288 is around 3 pg, and the other ORs detect Z5–14:OH in amounts >30 pg. Therefore, it is reasonable to conclude that Olfr288 is the most sensitive OR for Z5–14:OH.

**Preference for Z5–14:OH is mediated by Olfr288.** Next, we examined the involvement of Olfr288 in attractive behavior by performing the two-choice odor-preference test in WT and Olfr288-KO female mice. First, we performed a series of control experiments. Female mice spent similar time nose-poking two vents when the same urine from intact male urine was placed in both rooms (Fig. 3a). No difference in investigation time was observed between rooms when a neutral odorant, eugenol, was placed (Fig. 3a). In contrast, female mice were more attracted to urine from intact male mice (hereafter, intact urine) than urine from castrated males (hereafter, castrated urine) or water controls (Fig. 3a). In this experimental paradigm, aversion could not be measured because no difference was observed for TMT, a fox smell that causes avoidance behavior in mice (Fig. 3a). Number of poking noses into two vents, PG and TMT was also similar (Supplementary Fig. 4).

We performed the preference test for various amounts of Z5–14:OH (i.e., 10 pg, 1 ng, and 3 ng). WT female mice exhibited interest in 10 pg or 1 ng Z5–14:OH (Fig. 3b), but the attraction was diminished in Olfr288-KO mice (Fig. 3c). Similar results were obtained for castrated urine +/− Z5–14:OH. WT female mice were more attracted to Z5–14:OH (10 pg or 1 ng)-spiked

castrated urine than to castrated urine alone (Fig. 3d), as reported previously[9], whereas this preference was not observed in Olfr288-KO mice (Fig. 3e). Because 10 pg Z5–14:OH activates only Olfr288 (Fig. 2b), these results suggest that Olfr288 is involved in Z5–14:OH-mediated attraction in female mice.

In order to correct for any intrinsic special biases of individual mice, the "preference index" was calculated (Supplementary Fig. 5). Preference index means the ratio of investigation time of poking a nose into a hole with a targeted sample to that of the total investigation time into both holes. WT mice significantly preferred Z5–14:OH (10 pg or 1 ng)-spiked castrated urine in comparison to Olfr288-KO mice ($P = 0.0049$ for 10 pg, $P = 0.0031$ for 1 ng; Mann–Whitney test). There was no significant difference but a tendency for WT mice to prefer 10 pg and 1 ng Z5–14:OH in comparison to Olfr288-KO mice.

Interestingly, no preference was observed between water and 3 ng Z5–14:OH, nor between castrated urine and 3 ng Z5–14:OH-spiked castrated urine, in either WT or Olfr288-KO female mice (Fig. 3b–d). Because Olfr288-KO mice can still detect Z5–14:OH at amounts greater than 1 ng via the other Z5–14:OH ORs (Fig. 2b), we propose the following models; (1) 1 ng Z5–14:OH activates OlfrX(s), in addition to Olfr288; (2) 3 ng Z5–14:OH activates OlfrY(s), (one of the least sensitive OR among ~3 Z5–14:OH ORs); and (3) the activation of OlfrY(s) suppresses the Olfr288-mediated attractive behavior. In this experiment, however, we could not determine whether mice were simply not interested in 3 ng Z5–14:OH, or actively avoided it, because this nose-poking preference test cannot evaluate aversive behavior (Fig. 3a, TMT experiment). Number of poking noses into two vents was similar between water and 3 ng Z5–14:OH, or between castrated urine and 3 ng Z5–14:OH-spiked castrated urine, in either WT or Olfr288-KO female mice (Supplementary Fig. 4).

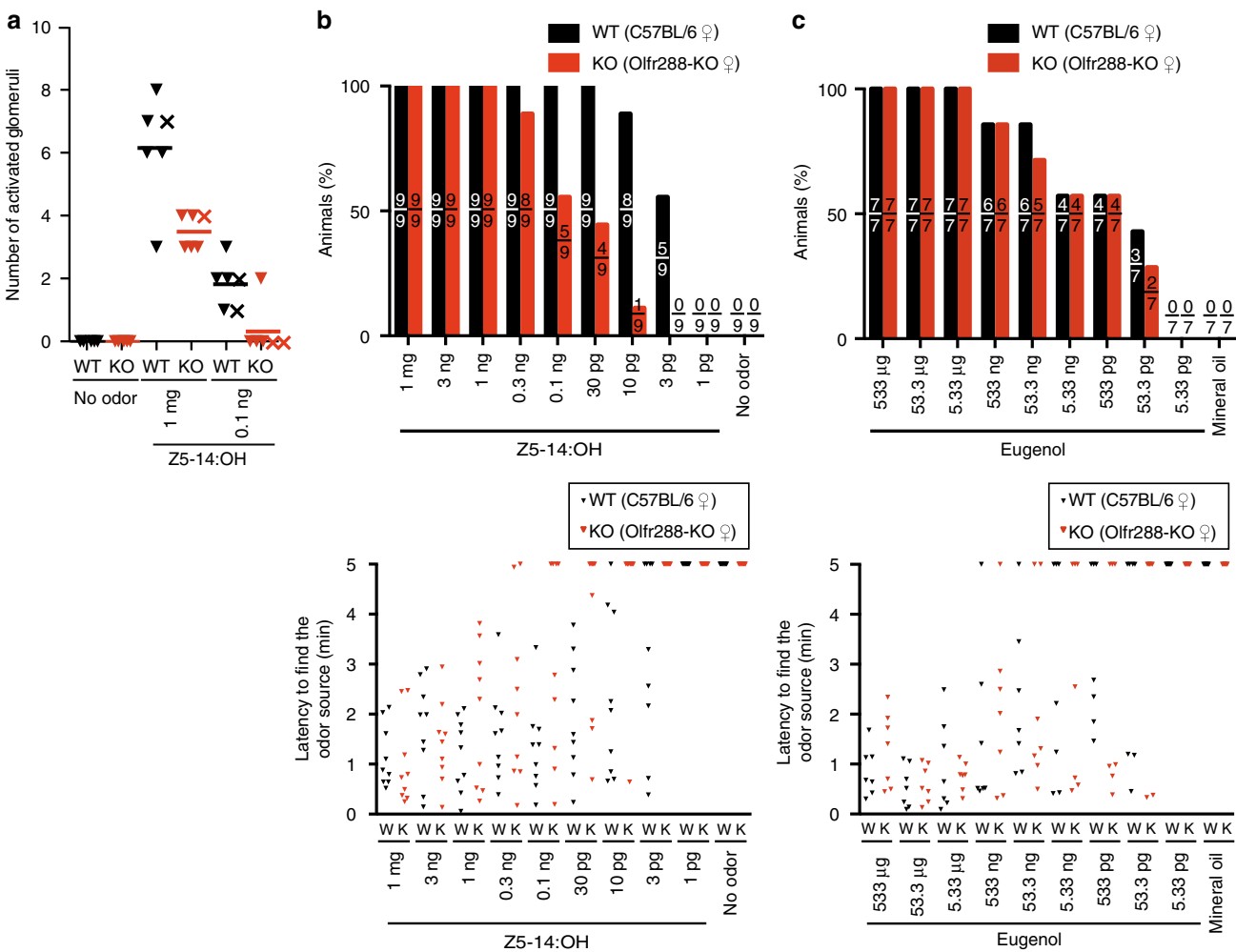

**Fig. 2** Presence of ~3 ORs for Z5–14:OH, including Olfr288, and the effect of Olfr288 deletion on detection of Z5–14:OH. **a** Number of activated glomeruli in the olfactory bulb of C57BL/6 adult female mice (WT) and Olfr288-KO adult female mice (KO) stimulated with 1 mg and 0.1 ng Z5–14:OH ($N = 6$) or without any stimulation ($N = 5$). c-Fos, an immediate early gene product, was used as an indicator of glomerular activation. Mann–Whitney test: 1 mg Z5–14:OH; $P = 0.0635$, 0.1 ng Z5–14:OH; $P = 0.1143$. Each triangle symbol depicts the result of immunohistochemistry and each cross symbol depicts the result of in situ hybridization. **b** Odor-finding test for Z5–14:OH in C57BL/6 adult female mice (WT) and Olfr288-KO adult female mice (KO) ($N = 9$). (Upper panel) The percentage of animals that found the odor sources within 5 min. The ratio of the number of animals that found the odor sources to the number of total animals that were tested is shown inside each bar. There were significant differences between WT and KO, as determined by the Scheirer–Ray–Hare test. **$P < 0.01$. (Lower panel) Latency to find the odor source in C57BL/6 mice (W) and Olfr288-KO mice (K). Each triangle depicts a single individual. Animals that failed to find Z5–14:OH in 5 min were plotted at 5. **c** Odor-finding test for eugenol in C57BL/6 adult female mice (WT) and Olfr288-KO adult female mice (KO) ($N = 7$). (Upper panel) The percentage of animals that found the odor sources within 5 min. There were no significant differences between WT and KO, as determined by the Scheirer–Ray–Hare test ($0.950 < P < 0.975$). (Lower panel) Latency to find the odor source in C57BL/6 mice (W) and Olfr288-KO mice (K). Each triangle depicts a single individual. Animals that failed to find eugenol in 5 min were plotted at 5

**Aversion to Z5–14:OH is mediated via the least sensitive Z5–14:OH OR.** To evaluate attraction and aversion in the same experiment, we performed an odor investigation assay in which a filter paper scented with an odorant was placed in the home cage, and the investigation time for the filter paper was measured during a 3-min test period (Fig. 4a, b). The investigation time for eugenol (EG), a neutral odor, was similar to that for water (Fig. 4c). Consistent with a previous report[25], in WT female mice the investigation time for TMT, an aversive odor, was significantly shorter (Fig. 4c). In addition, the investigation time for intact or castrated urine was significantly longer than for water control (Fig. 4c). These data demonstrate that both attractive and aversive behaviors can be evaluated in this investigation test.

We then investigated whether Z5–14:OH elicited attraction or aversion in this assay. The investigation time for 10 pg or 1 ng Z5–14:OH was significantly longer than that for water in WT

female mice (Fig. 4b, c), consistent with the results of the nose-poking preference test (Fig. 3). By contrast, the investigation time was significantly shorter for 3 ng Z5–14:OH than for water, suggesting that mice were aversive to 3 ng Z5–14:OH (Fig. 4c). The degree of aversion was similar to that of TMT. These results led us to propose that activation of the least sensitive Z5–14:OH OR evokes aversive behavior that overcomes the attraction mediated by Olfr288.

In Olfr288-KO female mice, behavior was similar for intact urine, castrated urine, EG, and TMT, whereas the attraction to 10 pg or 1 ng Z5–14:OH was completely diminished (Fig. 4c), consistent with the results obtained in Fig. 3. Further, the aversion to 3 ng Z5–14:OH was still observed in Olfr288-KO mice (Fig. 4c). All these results support our model that the attraction to Z5–14:OH is mediated via Olfr288, while aversion to a larger amount of Z5–14:OH is mediated by the least sensitive of ~3 Z5–14:OH ORs.

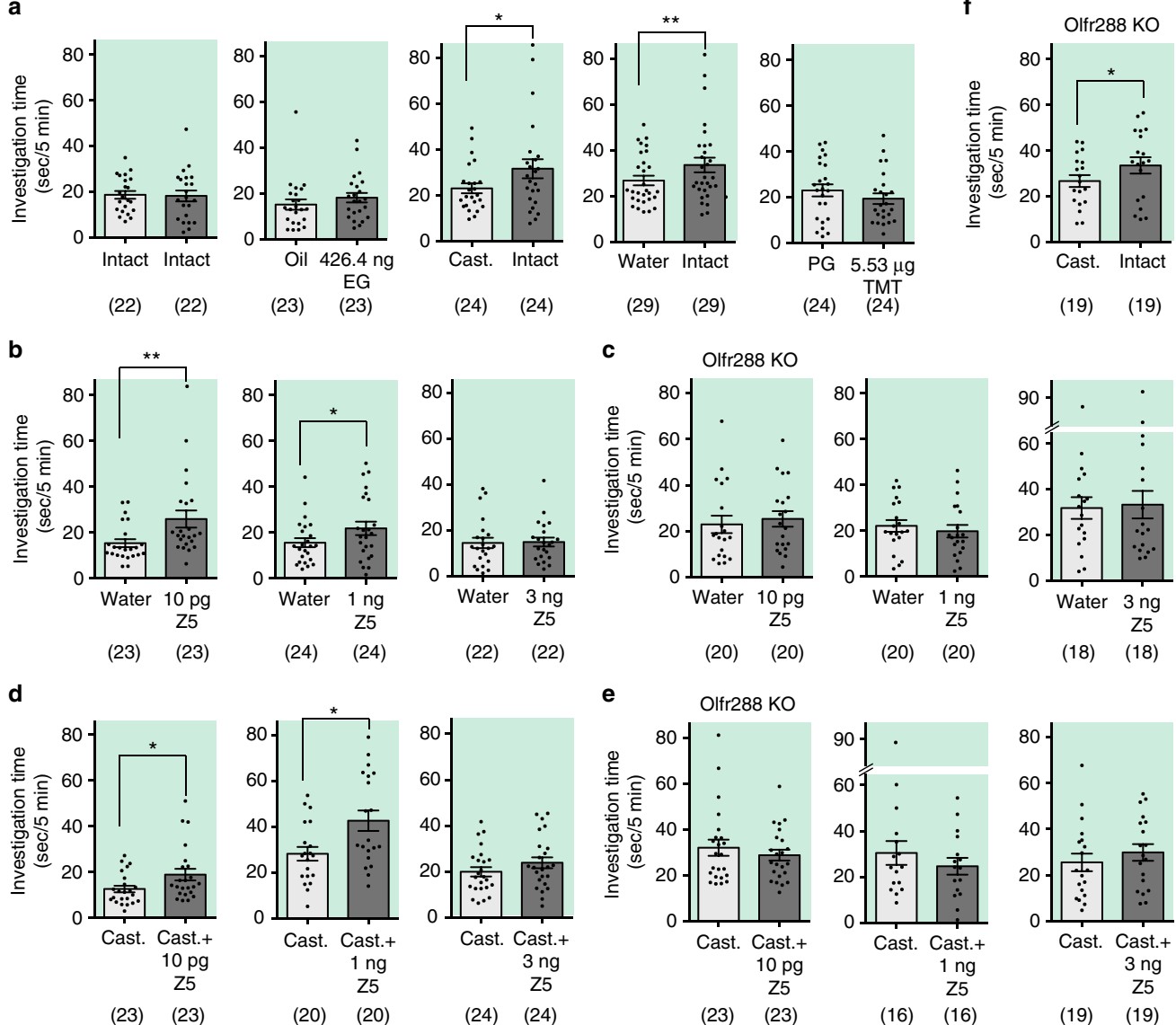

**Fig. 3** Attraction to Z5–14:OH is diminished in Olfr288 knockout mice. **a** Series of control experiments in the two-choice odor-preference test. Intact: urine from intact male C57BL/6 mice; Cast.: urine from castrated male C57BL/6 mice. TMT was diluted in PG. Oil: mineral oil; EG: eugenol. **b**, **c** Attraction to various amounts of Z5–14:OH of adult female C57BL/6 mice ($N = 22–24$) (**b**) and adult female Olfr288-KO mice ($N = 18–20$) (**c**). Z5: Z5–14:OH. **d**, **e** Preference for castrated male urine, with or without various amounts of Z5–14:OH, of adult female C57BL/6 mice ($N = 20–24$) (**d**) and adult female Olfr288-KO mice ($N = 16–23$) (**e**). **f** Preference for intact male urine vs. castrated male urine in adult female Olfr288-KO mice ($N = 19$). Bars indicate the mean time that a mouse spent investigating each test sample within a 3-min period, ±S.E.M. ($N = 16–29$). Asterisks indicate significant difference between two samples (paired Student's $t$-test, $^*P < 0.05$, $^{**}P < 0.01$)

## Discussion

Many odorants are neutral for mice (general odorants), whereas others possess specific valences such as attraction or aversion. The response is sometimes concentration-dependent, such that as the concentration increases, the output behavior shifts from attraction to aversion. An outstanding question is how odor valence is coded at the level of receptors. We can propose several models: the valence could be coded in the pattern of activated ORs, in individual OR(s), or by a combination of the two. In addition, valence may not be solely dependent on receptors, but also on the anatomy of activated regions in the olfactory bulb. In this study, to determine how behavioral output is regulated by ORs, we took an advantage of odorants that are recognized by only a few ORs, generated mice in which the most sensitive OR for a given odorant was deleted, and then performed behavioral analysis. Our results support the combination model in which activation of a single OR elicits preference or aversive behavior, and output behavior is determined by summation (addition or competition) of valences encoded by activated ORs.

We presented two cases: muscone and its two ORs, and Z5–14: OH and its ~3 ORs. In the case of muscone, as described in the bottom panels in Fig. 1c and d, MOR215–1 is likely to be involved in the attractive behavior, and MOR214–3 is an additional candidate OR that possesses the positive valence in MOR215–1 KO mice. In the case of Z5–14:OH, Fig. 5 shows a model that explains the mechanism underlying the concentration-dependent switch from attraction to aversion at the receptor level, and how individual Z5–14:OH ORs are involved in this behavior. There appear

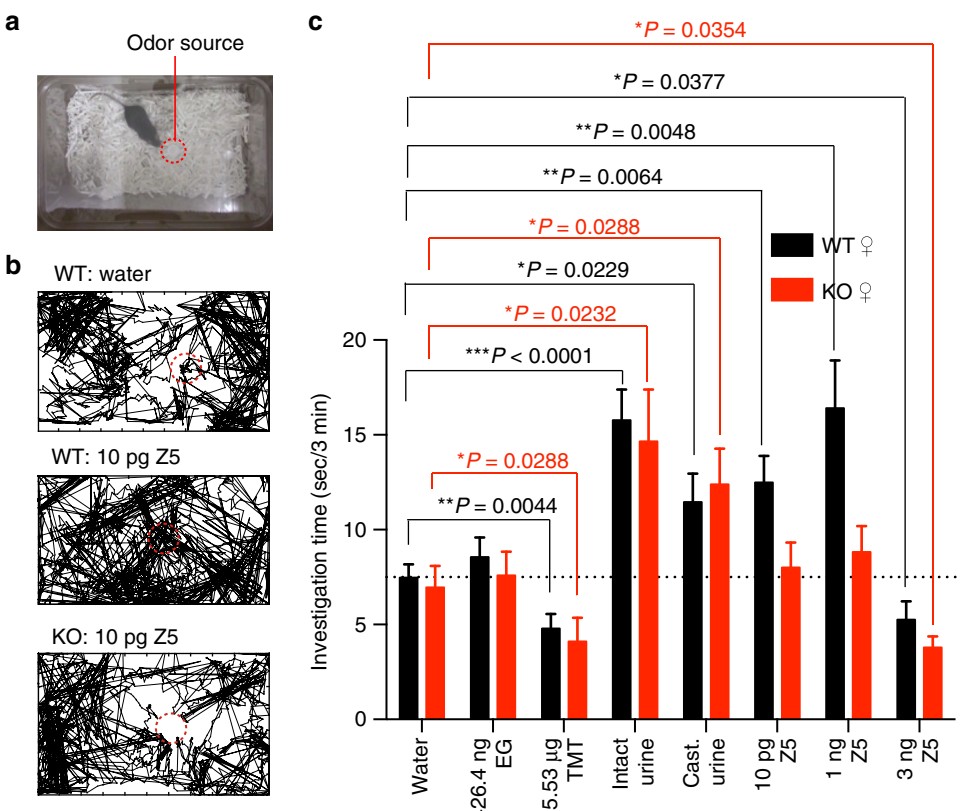

**Fig. 4** Aversion to high amounts of Z5–14:OH in both wild type and Olfr288 knockout mice. **a** Setup of the odor investigation assay. A mouse was exposed for 3 min to a filter paper bearing odorant placed in the middle of cage. The time the mouse spent investigating the odor source was measured. **b** Representative traces of wild type (WT) and Olfr288 knockout (KO) mice investigating 10 pg Z5–14:OH. **c** Odor investigation assay for various amounts of Z5–14:OH, intact or castrated urine, eugenol, or TMT, in adult female C57BL/6 mice (black bars) ($N = 15$) and adult female Olfr288-KO mice (red bars) ($N = 10$). Eugenol was diluted in mineral oil. TMT was diluted in PG. Z5: Z5–14:OH; oil: mineral oil; EG, eugenol; PG, propylene glycol. Data are means ± S.E. M. Mann–Whitney test: $*P < 0.05$, $**P < 0.01$, $***P < 0.001$

to be at least three ORs that recognize Z5–14:OH. The model most consistent with our results is that the most sensitive OR, Olfr288, encodes information about attraction, whereas the least sensitive (and still-unknown) ORs, a few OlfrY(s), sends the signal for aversion. The shift from attraction to aversion occurs when all ORs, including Olfr288, OlfrX(s), and OlfrY(s), for Z5–14:OH are activated. Taken together, our results suggest that each OR sends a signal to a neural circuitry leading to a specific behavior, and therefore encodes an odor-associate valence; the summation of valences coded by activated ORs determines the final output behavior.

To conclude that activation of a single OR is sufficient to induce a specific behavior, gain-of-function experiments will be needed, as reported previously for TMT[24]. Neurons expressing muscone receptors send axons to the dorsomedial part of the olfactory bulb. Glomeruli innervated with Z5–14:OH ORs are all located in the ventral region. Thus, accessing these bulbar regions are challenging upon applying optogenetic or pharmacological tools to activate individual muscone ORs or yet-identified OlfrXs or Ys. In addition, another challenge relates to the behavioral paradigm. Aversion can be assessed relatively easily by optogenetic or pharmacological stimulation[24], whereas preference behavior is more difficult to evaluate. A more refined behavioral paradigm must be developed for assessment of positive valences. In addition, a stereotyped neural circuit transmits information from the olfactory bulb to cortical amygdala, resulting in various innate behaviors. Distinct cell populations within the cortical amygdala are capable of eliciting innate responses to either preference or aversive odorants[21]. It remains to be determined how

the behavioral outputs of muscone and Z5–14:OH are correlated with activation patterns in the higher brain areas, especially the cortical amygdala.

Based on our findings, we propose that some ORs code a valence such as information about preference or aversive behavior, thus linking individual ORs are with specific behavioral outputs. The final output behavior is determined by addition/ competition of valences coded by activated ORs, suggesting that convergence and cross talk among stimuli from multiple ORs occur within the brain. From the standpoint of applications, the concept of OR-associated negative or positive valence could be exploited to regulate animal behavior, e.g., controlling a pest or a harmful animal by development of agonists or antagonists for the target OR. In human society, this concept could be applied to design of improved fragrances or flavors by targeting an essential OR involved in detecting a specific odorant quality.

## Methods

**Odorants.** Muscone and mineral oil were purchased from Wako. Propylene glycol (PG) was purchased from ADEKA. (Z)−5-tetradecen-1-ol (Z5–14:OH) (>90%; including 7.1% of the E-isomer) was synthesized by Chegenesis Co. via a synthetic pathway described previously[26]. Eugenol (EG) and TMT were purchased from Tokyo Chemical Industry Co. or Phero Tech.

**Animals.** All experiments were carried out in accordance with the guidelines of the Animal Care Committee at The University of Tokyo. The animal room was maintained under a 12-h light/dark cycle (light from 7:00 to 19:00) at a constant temperature (23 ± 1 °C). C57BL/6 male and female mice (>8 weeks old; CLEA Japan) were used for all experiments. MOR215–1 deletion mice were obtained from Dr. Sakano[27]. The coding sequence of *MOR215–1* was changed to that of *MOR103–1* using recombinant PCR in the *MOR103–1->MOR215–1 IRES tau-*

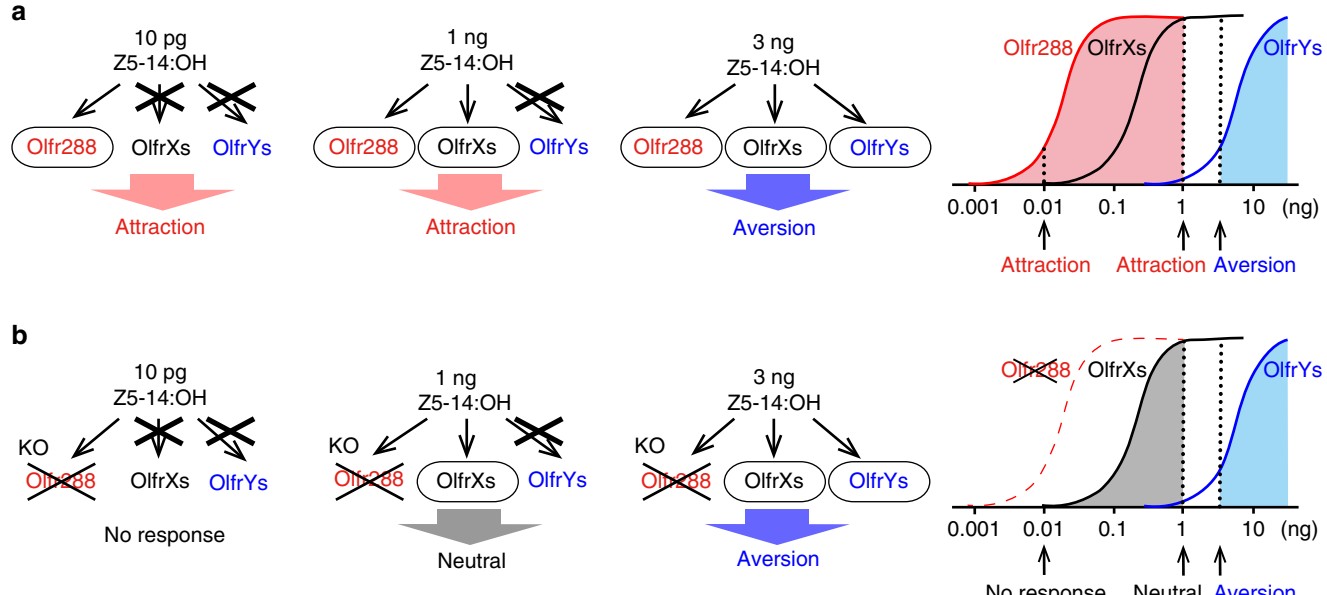

**Fig. 5** A receptor-based model for concentration-dependent switching from attraction to aversion to Z5–14:OH. Behavioral data of wild-type mice (**a**) and Olfr288 knockout mice (**b**) in response to various amounts of Z5–14:OH (10 pg, 1, 3 ng) are consistent with a model (**a**, **b** right) in which activation of Olfr288 (red) and OlfrY(s) (blue) are involved in attraction and aversion to Z5–14:OH, respectively; activation of OlfrX(s) are neutral

*ECFP* mouse (Acc. no. CDB0545K; http://www.cdb.riken.jp/arg/mutant%20mice%20list.html). Olfr288 deletion mice were generated as follows. A genomic fragment of mouse *Olfr288* was isolated from the B6N Mouse BAC clone B6Ng01–336J02 (*RIKEN* BioResource Center). A 2.5-kb fragment spanning the 5′ region upstream of the third exon was amplified by PCR with KOD FX (Toyobo) and inserted into the pBluescript II SK(−). A self-excision Neo^R cassette derived from pACN[28] and a 6.3-kb fragment of *Olfr288* genomic sequence spanning from upstream of the second exon to downstream of first exon was then inserted into the 3′ region of the Neo^R cassette. To allow negative selection, a diphtheria toxin A fragment gene cassette derived from pMC1DTpA[29] was inserted. The targeting vector was linearized with *Not*I and electroporated into C57BL/6 mouse ES cells. Cells that underwent recombination were selected with G418, and colonies were screened by PCR. Male chimaeras were crossed to wild-type C57BL/6 females (Japan SLC) to establish a C57BL/6 inbred background. The mutant mice were analyzed by PCR using two primer pairs (5′-tgagatggctcagcaggtaagag-3′ and 5′-gacagttacgatact ctcctgg-3′ for the wild-type allele, and 5′-cgaagttataagctttcgcgagctc-3′ and 5′-gacagttacgatactctcctgg-3′ for the mutated allele).

**Two-choice odor-preference test**. Two-choice odor-preference tests were performed in a custom-made acrylic box (300 mm × 210 mm × 100 mm) consisting of three compartments: one mouse compartment (155 mm × 210 mm) with clean bedding, and two odorant compartments (143 mm × 104 mm), as described previously[9]. The tests were conducted during the late part of the light period, and were performed in a double-blind manner. Each adult mouse was habituated for 15 min before each test. Each odorant (5 μl) was applied to a piece of filter paper (15 mm diameter) on a dish and placed in an odorant compartment; the positions of the odorants were counterbalanced among tests. The partition between the mouse compartment and each odorant compartment contained a round vent through which air could pass or the mouse could poke its nose. A stream of air was drawn through a charcoal filter from each odorant compartment into the mouse compartment passing along a Y-shaped tube using a vacuum pump. Mice were recorded with a digital video camera. Each test was performed for 5 min, and the amount of time a mouse spent poking the tip of its nose through the vent, designated as the investigation time, was measured by using a stopwatch in a double-blind manner. The investigation time was compared statistically between odorant compartments by paired Student's *t*-test. To avoid confounding of the data due to learning, each mouse was used only once.

**c-Fos immunostaining**. Mice were housed individually with clean bedding. After 2 days of isolation, the tip of a glass capillary with each odorant was set carefully in the home cage. The glass capillary was kept in the home cage for 1 min after freely behaving mice found and began sniffing the odorant. Then the glass capillary with odorant was removed from the home cage carefully. As a control, mice were stimulated through a glass capillary with no odorant for 1 min. After 70 min, mice were anesthetized with sodium pentobarbital and perfused with PBS and 4% paraformaldehyde in PBS at 4 °C. The skull was removed so that the olfactory bulb

(OB) was exposed, and the OB was post-fixed for 3 h and stored overnight in cryoprotection solution (30% sucrose in PBS). Thirty-micron coronal cryosections of the OB were collected, placed onto MAS-coated glass slides (Matsunami Glass), and incubated with anti-c-Fos rabbit polyclonal antibody (1:500, sc-52, Santa Cruz Biotechnology). The slides were then incubated with biotinylated goat anti-rabbit secondary antibody (1:200, PK-6101, Vector Laboratories), subjected to ABC amplification (Vector Laboratories), and stained with 3,3′ diaminobenzidine (Sigma). Number of activated glomeruli that expressed c-Fos in juxtaglomerular cells were counted from one side of the OB from each mouse. No c-Fos induction was observed in the OB of mice with no odor as a control.

**Odor-finding test**. Each mouse was housed individually in its home cage with clean bedding. After 2 days of isolation, the odor-finding test was performed as described previously[5,15] in the same cage. The mouse was exposed to the tip of a glass capillary with an odorant. The tip of each capillary was set carefully so that the mouse could not reach it directly, and the latency time before the mouse began actively sniffing the odorant (i.e., the tip of the capillary) was recorded. Initially, the mice were exposed to a low concentration of the odorant. If a mouse did not find the odorant, the concentration of the odorant was increased. The experiment was performed on each mouse until it found the odorant, or up to three times a day. Mice were recorded with a digital video camera. Experiments were conducted during the dark periods, and were performed in a double-blind manner. The latency of finding the odorant was compared statistically between WT and KO mice by the Scheirer–Ray–Hare test, a nonparametric version of repeated two-way ANOVA[30,31].

**Odor investigation assay**. Mice were housed individually in their home cages with clean bedding. After 1 day of isolation, the odor investigation assay was performed as described previously, with minor modifications[25], in the same cage. A filter paper (15 mm diameter) scented with an odorant (5 μl) was introduced to the home cage, and the time spent investigating the filter paper during a 3-min test period was measured. Mouse behavior was recorded with a digital video camera for analysis. To avoid confounding of the data due to learning, each mouse was used only once. Tests were conducted during the late part of the light periods, and were performed in a double-blind manner. The investigation time was compared statistically between mice exposed to control (water) and test odorant using the Mann–Whitney test. The dashed line in Fig. 4c means the average time of water in WT mice.

**RT-PCR**. Olfactory sensory neurons were isolated from olfactory epithelium and RNAs were purified by RNeasy Plus Mini Kit (Qiagen). 1000 ng RNAs were each placed into lysis mix (0.5 μl Oligo dT (invitrogen), 2.5 mM dNTPs (Takara)) and RNase free water was added till 14 μl. PCR tubes each containing lysed cells were heated to 65 °C for 5 min and then cooled at 4 °C. 6 μl of RT-positive mix (1 μl Superscript III (Invitrogen), 1 μl RNase inhibitor (Takara), 4 μl 5 × FS buffer

(Invitrogen) and 1 μl 100 mM DTT (Invitrogen)) was added to each reaction. As a control, RT-negative mix without Superscript III was made. The mixtures were incubated at 50 °C for 60 min and then 70 °C for 5 min. Next, 2 U/μl RNAse H (Roche) was added to each tube and tubes were incubated at 37 °C for 30 min and then 29 μl water was added. The cDNA generated from each sample was PCR-amplified using the following primers: For Olfr288; (Primer-F) AGACAAGTCT TTCTCCCTCCTCTATAC, (Primer-R) ATTCCTAAATGTATGTGCAAAAAGT TC, For β-actin; (Primer-F) TTGTAACCAACTGGGACGATATGG (Primer-R) GATCTTGATCTTCATGGTGCTAGG. Samples (1 μl) of the products of each reaction were each added to 49 μl of PCR mix (1 × ExTaq buffer (Takara), 0.25 mM dNTPs, 1 μM primers (Primer-F and Primer-R), 1.25 U Ex Taq HS polymerase (Takara)) and incubated at 95 °C for 15 min, and then 35 cycles of 94 °C for 30 s, 58 °C for 30 min, and 72 °C for 30 s for each cycle, then 72 °C for 10 min. After amplification, reaction solutions were subjected to 1.5% agarose gel electrophoresis with ethidium bromide. RT-negative reactions were run as a negative control to distinguish PCR products from mRNA and genomic DNA. β-actin was used as internal control.

**In situ hybridization.** To make the Olfr288 gene probes, DNA fragments of 371 bp containing the 3' untranslated region (UTR) sequences were amplified by PCR from the C57BL/6 mouse genomic DNA. Primer sequences for Olfr288 were AGAAGGAGGGTGTCTGGAGAAG (forward) and AATTAACCCTCACTAAA GGGTCCAATACCAAGTCTAATATGGC (reverse, with T3-sequence at 5' end). To make the c-Fos gene probes, DNA fragments of 888 and 905 bp containing the coding sequence were amplified by PCR from the C57BL/6 mouse cDNA from the brain. Primer sequences for c-Fos were CCAGCTCTGCTCTGCAGCTC and CGGTTCCTTCTATGCAGCAG (forward) and AATTAACCCTCACTAAAG GGGAAGTCATCAAAGGGTTCTG and AATTAACCCTCACTAAAGGGCA CAATAAAAACGTTTTCATGG (reverse, with T3-sequence at 5' end). Antisense cRNA probes were synthesized using T3 RNA polymerase (NEB) and digoxigenin (DIG) labeling mix (Roche) from PCR templates. Thirty-week-old C57BL/6 and Olfr288-KO mice were anesthetized with sodium pentobarbital (2.5 mg/animal) and perfused intracardially with 4% paraformaldehyde. Olfactory tissues were dissected out and fixed overnight in 4% paraformaldehyde in PBS. Tissues were decalcified by incubation for 4 days in 0.5 M EDTA at 4 °C, placed in 30% sucrose. Fifteen μm coronal sections of olfactory epithelium were prepared. After drying, the samples were fixed for 10 min in 4% paraformaldehyde in PBS at room temperature. The sections were rinsed with PBS and incubated with 1 μg/ml Proteinase K in PBS for 5 min at 37 °C. After fixing again with 4% paraformaldehyde in PBS for 10 min and rinsing with PBS, the sections were incubated with 0.1 M triethanolamine with 2.5 μl/ml acetic anhydride, pH 8.0, washed with PBS, and the slide was incubated with the hybridization solution (50% formamide, 5 × SSC, 1 × Denhardt's, 250 μg/ml yeast tRNA) for 10 min. Probes were diluted (1:200) with the hybridization solution, and 300 μl of each sample was applied to a slide. After 12 h of incubation at 68 °C, the sections were washed, first with 5 × SSC for 5 min, then with 0.2 × SSC three times for 30 min at 68 °C. After blocking with the blocking reagent (PerkinElmer), slides were incubated with anti-DIG-POD (Roche, at 1/1000 dilution in blocking reagent) for 3 h at room temperature. Slides were washed three times with TBST (100 mM Tris, pH 7.5, 150 mM NaCl, 0.05% Tween 20) for 10 min and treated with TSA-plus Cyanine 3 (PerkinElmer, NEL744001KT, 1:100 in 1× plus amplification diluent) for 20 min. Sections were washed three times with TBST for 10 min and mounted with cover glass using PermaFluor (Lab Vision Corporation), and imaged by Keyence microscope (×4 or ×10 objective). Images were processed in Photoshop CC (Adobe).

## Data availability
The relevant data that support the findings in this study are available from the corresponding author on request.

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

## Acknowledgements
We thank members of the Touhara lab for helps and valuable discussions. This work was supported in part by Grant-in-Aid for Scientific Research (S) from MEXT Japan (Grant Number 24227003) and ERATO Touhara Chemosensory Signal Project (JPMJEN1202) from JST, Japan. N.H. is a recipient of Grant-in-Aid for JSPS Research Fellow from JSPS Japan (Grant Number 12J08216) and Grant-in-Aid for Young Scientists (B) from MEXT Japan (Grant Number 26850061).

## Author contributions
N.H. and K.T. conceived the project, designed the experiments, and wrote the manuscript. N.H. performed all experiments except that K.M. performed expression analysis of

knockout mice. K.Y. performed an initial experiment related to Fig. 3. Y.Y. helped generation of knockout mice.

## Additional information

**Competing interests:** The authors declare no competing interests.

