## [Peer Review File · Nature Communications]

Reviewers' Comments:

Reviewer #1:

Remarks to the Author:

The manuscript by Horio et al. describes experiments investigating the role of individual odorant receptors on animal behavior, focusing on two previously-described ligands that have only a handful of known high-affinity receptors. The strength of the manuscript is its focus on behavioral valence and the large cohorts of animals studied in these conditions. However, several weaknesses prevent enthusiasm for the work in its current form. First, the muscone valence studies (Fig. 1) provide only minimal new information about the relationship between muscone, its receptors, and animal behavior. Importantly, the conclusions reached by the authors regarding the mechanisms by which the Z5 odorant-receptor interactions lead to the observed behaviors are not directly supported by the data. Though the topic is interesting, as presented here the data remain too superficial to support the major conclusions.

Major points:

1) Previous work by the laboratory (Sato-Akuhara et al, 2016) identified the principal mouse receptors for muscone and established the principal effects of MOR215-1 knockout on muscone perception. The results of the muscone preference/attraction assay here suggest that MOR214-3 may provide sensitivity/positive valence at higher muscone concentrations, but without MOR214-3 knockout/manipulation on top of the MOR215-1 knockout, this conclusion is only indirectly supported. Without such a direct test, it is not clear whether other receptors besides those currently known for their muscone sensitivity are not at play.

2) The data in Figure 2A (counting olfactory bulb Fos staining, no raw data shown) are not subjected to statistical analysis. Combined, these two factors render the data uninterpretable and unworthy of inclusion in this form. The perception test applied in Fig. 2B is more informative. The authors could strengthen their claim that Olfr288-KO female mice are selectively deficient in their ability to perceive low concentrations of the Z5 ligand by adding Fos+ glomerular counting in a cohort of WT and Olfr288-KO mice that were only exposed to the low concentrations of Z5 (e.g. 10 pg).

3) Though this point is noted by the authors, the lack of the ability to observe aversive responses to TMT in the two-odor preference assay is alarming. Do the animals not freeze or display stereotypical fear-associated behaviors in these conditions? If not, this casts doubt on the efficacy of the paradigm in general. It would be useful to provide data (e.g. from a photoionization detector or other quantitative measurement) indicating the spatial extent and effective concentrations of odorants flowing through their setup.

4) The results of the home cage aversion assay are interesting, and do reveal an important switch in valence at higher concentrations of the Z5 ligand. However, the model put forth in Figure 5, which is a major focus of the manuscript, remains entirely speculative at this time. Are there indeed two additional receptors that can account for this switch? Five? Without any additional evidence the model lacks compelling support.

Minor points:

Results, p. 8. Second to last paragraph (related to Figure 2A): no statistical comparison is listed in text or figure/legend. Given the low N and apparent variance in the measurements, this data is not compelling

Discussion, p. 11, line 13. "muscone and its two ORs" and "Z5-14:OH and its three ORs" – it would be appropriate to clarify each by saying "strongest ORs"

Language/typographical:

Intro, p.4, line 4, "is not practical" – redundant phrase

Methods, p. 6, line 2-3. "A stream of air was drawn through the charcoal filter from each odorant."
Confusing. Please clarify

Figure 4D (figure text): "attracton" should be "attraction"

Reviewer #2:

Remarks to the Author:

In this manuscript Horio et al. attempt to uncover the contribution of individual olfactory receptors to naturalistic behaviors. The most interesting finding is that removing the activity of one OR species from an ensemble of odor-activated ORs has the potential to drastically change odor perception and subsequently behavioral output. The authors use innate odor-driven behaviors as an assay of perception and preference. This approach has several advantages over learned behaviors, but also requires some careful controls. These findings of this study may be novel and potentially impactful; however, there are several issues that confound the interpretation of the experimental results.

Major concerns:

Figure 1:

1) The methods section is lacking clear details on how the nose poke was measured and quantified. Was the data obtained visually, using a photodiode, or by automated detection from images? It would be helpful to know whether the animal was required to place its nose through the port or only investigate the immediate vicinity.

2) Was air blown through each chamber prior to odor being added? Furthermore, was the flow rate from each odor chamber measured and equilibrated? This is especially important if the mouse was required to place his nose through the port as discussed in the previous point.

Figure 2:

1) Further characterization of the Olfr288-ko mouse is necessary. This seems to be a previously unpublished mouse line and there is no supporting evidence to demonstrate that olfr-288 is absent from the olfactory epithelium, as well as the in the olfactory bulb itself. An in-situ image would be helpful.

2) "freely behaving mice were stimulated through a glass capillary with each odorant for 1 min after they found the odor source." - this experimental description lacks key details. It is unclear to this reader how the mice were stimulated with the odor. Self-administration vs forced administration of the odor may result in differences in cFos expression.

3) Some example images of cFos staining are necessary. Further, how activated glomeruli were detected and quantified is missing from the methods. There is no discussion of how active glomeruli were detected over those that are activated by background odors in the home cage. Some measurements of the relative positions of the active glomeruli would certainly be helpful to provide some assurance the same ORs are activated in different bulbs/animals.

4) Was the number of activated glomeruli counted in each olfactory bulb, or does this number consider the total across both bulbs? If the value is from a single olfactory bulb, was more than one bulb from each mouse considered in the analysis?

5) Perhaps a more sensitive analysis to consider for the odor-investigation (also the choice task) assay is the latency to find the odor source. For both the WT and KO mice the ability to find the odor source saturates around 1 ng. By measuring the latency to find the odor some deficits may be observed in KO mice. The methods state that this was how the experiment was performed and analyzed, but the data displayed in Figure 2B only present the binary data of success or failure to locate the odor source.

Figure 3:

1) The results of the eugenol experiment are surprising. Mice seek novelty and the presence of a neutral odor should elicit some investigation.

2) No aversion is detected to TMT; however, by comparing the number of nose pokes to a period of time prior to the odor delivery, a decrease may be observed. The data appear to be available as there was a 15 minute habituation period for each mouse. This comparison could also help resolve whether a 3 ng concentration of Z5 is aversive. By comparing the relative number of nose pokes prior to and following odor delivery, the authors can correct for any intrinsic spatial biases of individual mice.

3) It is unaddressed as to whether there is a possible trigeminal component to the detection of Z5. In other words, at sufficiently high concentration the odor vapor could generate sensation in the nasal cavity and the mucosa around the eyes. In this scenario, the switch from attraction to aversion could be the result of non-olfactory cues contributing to behavioral output.

Figure 4:

1) Displaying the position of the nose rather than the center of mass would be beneficial. As displayed it appears that the example mouse avoids, rather than investigates the odor source.

Minor concerns:

1) Introduction line 4: Olfactory neurons should be changed to olfactory sensory neurons or receptor neurons.

2) Introduction paragraph 3: Olfactory neural pathway is ambiguous - this should be explained in some detail.

4) Introduction last paragraph, sentence 2: "is not practical" can be removed from the end of the phrase.

5) Methods, Two-choice odor-preference test, sentence 1: Acryl should be changed to acrylic

3) For each bar graph, showing the distribution of individual points is preferred over simple bars with error.

Reviewer #3:

Remarks to the Author:

The paper by Dr. Touhara and colleagues addresses an important problem in the field of olfaction: Do individual sensory neurons encode valence? A growing body of work suggests that indeed some olfactory sensory neuron (OSN) types are wired to elicit behavior of valence. In the present study, the authors use mouse olfactory receptor deletions to partially deconstruct the contribution of single OSN types to attraction to odors, with presumed innate social significance. Specifically, they find that MOR215-1 is necessary for attraction to low levels of muscone, but not high amounts. By inference they conclude that this attraction to high intensity odor is due to the activation of MOR214-3, but this is speculation. Similarly, they demonstrate that Olfr288 is necessary for detection of low levels of the odorant, Z5-14-OH. Again by inference, they conclude that an unidentified OSN type plays a role in attraction to moderate levels of the odorant, and another OSN type shifts the behavior to aversion at high concentrations—speculation, but probable. Overall, the authors address an important question, but there are concerning gaps in the relationship between OSN activation and behavior. Moreover, the behavioral data is not particularly convincing.

I have the following specific concerns.

1. Throughout the paper, the behavioral assays performed are not particularly sophisticated or sensitive, and seem to be quite variable. This makes the results less convincing and I have doubts about the reproducibility of the findings. For example, in Figure 1, the authors use a two-choice assay to compare the amount of time in a port with control air, propylene glycol, to the time in a port with muscone. The amount of time they spend in the control port ranges from about 11-24 sec, whereas the difference between the two ports is a maximum of about 10 sec, which is within the range of variation. Further, the "significant" result in 1D is so unremarkable that it would go unnoticed if not for the asterisk. What are the standard deviations in these measurements? To their credit, the experiments appear to be performed in a double-blind manner.

2. The authors infer the role of untested OSN types in attraction and avoidance behavior, but the conclusions are too strong in the absence of direct evidence. For example, the authors conclude that the attraction to high levels of muscone is due to the activation of MOR214-3 without directly testing this. The inference is made based on past characterization of musk odor receptors in vitro with heterologous expression. However, it remains possible that another OSN type contributes to the behavior; the past citation is not sufficient to conclude that no other OSNs are activated by the odor. In the case of the elusive OlfrX & OlfrY, the authors admittedly don't know the identify of the odorant receptors. Although I agree that the behavioral data is consistent with a model implicating at least two other OSN types, this is speculative and the conclusions should be tempered.

3. Fig 2A: The authors use c-fos to assess the number of activated glomeruli by the odorant in wt and Olfr288 KO mice. First, they should show some histology for verification of the c-fos detection and evaluation of glomerular identification. Second, it is peculiar that they do this comparison only at a high concentration of odorant, in which there is no behavioral difference between the groups (Fig 2B). Correlating behavior with glomerular activation should be done at the same concentration. Indeed, a careful c-fos analysis of glomerular activation would strengthen their argument about the number of

glomeruli or OSN types involved at different concentrations throughout the paper.

4. Fig 3 B-E: The statistical analysis is inadequate to support the conclusion that there is a difference between wt and Olf288 KO mice. The authors employ a paired T-test to compare investigation time for each group of mice exposed to two odors, which is appropriate to say that a given group prefers one odor over the other. However, there is no statistical comparison between groups. The authors conclude that Olfr288 is required for attraction to the odorant, but this conclusion must be supported by a statistical analysis comparing the two groups. This seems especially important given that the Olfr288 KO mice investigate the odor about as much as the wt mice.

5. Fig1, 3,& 4: The axis labels, "Investigation time (sec/3 min)," are confusing. Are the data presented as a fraction of total time or as total seconds? The legends state total time, which is in contrast to the axis label.

6. Fig 4C: The bottom axis has the label, "milli Q." What is this? I presumed it was water, but this should be clarified in the figure or legend. Additionally, the dashed line should be defined in the legend.

7. The MOR215-1 KO mouse provided by Dr. Sakano is cited, but the cited paper does not have this KO. The paper does have a mouse in which MOR215-1 is replaced by MOR103-1. Is this the mouse used in the current manuscript? Although, technically a deletion, this is more complicated and should be clarified in the manuscript.

8. There is an inconsistency between the methods and the figures for the two-choice odor-preference test. Methods state that the test was performed for 5 min whereas the figures state 3 min. This should be clarified.

9. In the results section and Fig 5B, the authors state that the Olfr288 KO mice cannot detect odor at 10 pg or less. However, the assay employed cannot distinguish between cannot detect and not interested in the odor. It would be more appropriate to state that mice do not respond, than do not detect.

Reviewer #1 (Remarks to the Author):

Major points:

1) Previous work by the laboratory (Sato-Akuhara et al, 2016) identified the principal mouse receptors for muscone and established the principal effects of MOR215-1 knockout on muscone perception. The results of the muscone preference/attraction assay here suggest that MOR214-3 may provide sensitivity/positive valence at higher muscone concentrations, but without MOR214-3 knockout/manipulation on top of the MOR215-1 knockout, this conclusion is only indirectly supported. Without such a direct test, it is not clear whether other receptors besides those currently known for their muscone sensitivity are not at play.

RESPONSE:

We agree that without MOR214-3 knockout/manipulation on top of the MOR215-1 knockout, we cannot conclude that MOR214-3 provides sensitivity/positive valence at higher muscone concentrations. We are sorry that our conclusion was an overstatement. It, however, takes a few years to generate and analyze MOR215-1/MOR214-3 double knockout mice, and even if we did that, it is still possible that the preference behavior does not disappear because it is possible that there is an unidentified muscone OR involved in the attractive behavior. Nonetheless, it is fair to say that regardless of the phenotype of the double knock-out mice, our conclusion that in addition to MOR215-1, there is other OR(s) involved in attraction to muscone is the same. Therefore, we toned down the sentence in Results “Together, these results suggest that both MOR215-1 and MOR214-3 are involved in attraction to muscone (Figure 1C and D, lower panels).” to “Together, these results suggest that MOR215-1 and other receptor(s), likely MOR214-3, are involved in attraction to muscone (Figure 1C and D, lower panels)” (page 9, line 20-21). Further, we clearly indicate that the schematic diagrams in Figure 1C and D, lower panels, are just a model by saying “A possible model” in the legend. We also toned down the sentences in Discussion; “Both MOR215-1 and MOR214-3 are likely to be individually involved in the attractive behavior.” to “MOR215-1 is likely to be involved in the attractive behavior, and MOR214-3 is an additional candidate OR that possesses the positive valence in MOR215-1 KO mice” (page 13, line 16-18). The sentences in Abstract was also toned down; “Muscone is recognized by two ORs, MOR215-1 and MOR214-3, and attracts male mice. Odor preference tests using MOR215-1 knockout mice revealed that both MOR215-1 and MOR214-3 are involved in attraction to muscone.” to “Muscone is recognized by a few ORs including MOR215-1 and MOR214-3, and attracts male mice. Odor preference tests using MOR215-1 knockout mice revealed that both MOR215-1 and other OR(s), likely MOR214-3, are involved in attraction to muscone.” (page 2, line 6-9)

2) The data in Figure 2A (counting olfactory bulb Fos staining, no raw data shown) are not subjected to statistical analysis. Combined, these two factors render the data uninterpretable and unworthy of inclusion in this form. The perception test applied in Fig. 2B is more informative. The authors could strengthen their claim that *Olf288*-KO female mice are selectively deficient in their ability to perceive low concentrations of the Z5 ligand by adding Fos⁺ glomerular counting in a cohort of WT and *Olf288*-KO mice that were only exposed to the low concentrations of Z5 (e.g. 10 pg).

RESPONSE:

We are sorry for not performing statistics nor showing raw data. Regarding the statistics, we did the Mann-Whitney test and found that the P value between WT and KO to 1 mg and 0.1 ng Z5-14:OH was P=0.0635, and P=0.1143 each. This was added in the figure legend in the revised manuscript. Regarding the raw data of c-Fos staining, we now show them in Supplementary Figure 2 (Figure R1 below). The arrowheads indicate the activated glomeruli. The relative positions of the active glomeruli in the olfactory bulb were shown in Supplementary Figure 3 (Figure R2 below). Our data suggest that the same ORs are likely activated in different bulbs/animals.

We did the c-Fos staining at a lower concentration. When mice were stimulated with 0.1 ng Z5:14:OH, the number of activated glomeruli in WT and KO was ~2 and ~0, respectively (Figure 2A, and Figure R3 below). We should note that the threshold for the KO mice in the odor-finding behavioral test (0.01 ng, Figure 2B) was lower by an order of magnitude. This threshold difference (0.1 ng in c-Fos and 0.01 ng in behavior) could be explained by the previous observation that the threshold of sensitivity in the olfactory bulb imaging is higher than that in the odor-finding behavioral test (Oka et al., *Neuron*, 52:857–869, 2006; Sato-Akuhara et al., *J Neurosci*, 36:4482–4491, 2016). Nonetheless, we can conclude that the *Olf288*-KO mice are selectively deficient in perceiving low concentrations. We added these data in Figure 2A, and Supplementary Figure 2B.

Figure R1

Figure R2

Figure R3

3) Though this point is noted by the authors, the lack of the ability to observe aversive responses to TMT in the two-odor preference assay is alarming. Do the animals not freeze or display stereotypical fear-associated behaviors in these conditions? If not, this casts doubt on the efficacy of the paradigm in general. It would be useful to provide data (e.g. from a photoionization detector or other quantitative measurement) indicating the spatial extent and effective concentrations of odorants flowing through their setup.

RESPONSE:

We measured the immobility time for TMT using the same video data in Figure 3, and we could not see the significant difference between TMT and a control odor, eugenol (EG), in the immobility time (Figure R4 below, Mann-Whitney test; $P=0.9538$). Unfortunately, we could not measure exact special concentrations of odorants, but we can argue the reason for the failure to observe the aversive/immobility behavior in the two-choice odor-preference test as follows. To see immobility and aversive behavior, Saito et al. (Nature Communi. 8:16011, 2017) utilized 100 μl of 10% TMT in the open-field assay, whereas one-tenth amount of TMT caused much weaker aversive behavior. This amount of TMT in Saito et al. is much larger than that used in our two-choice odor-preference test (5 μl of 0.1% TMT) (Figure 3A). Thus, although we cannot exactly compare their study and ours, it is possible that the amount of TMT utilized in the two-choice odor-preference test might be too small to see the aversive/immobility behavior. Another possibility is that the interest to poke a nose into a hole overcame the behavior to avoid TMT odor. In contrast, using the similar method as our odor investigation test in Figure 4, Kobayakawa et al., (Nature, 450:503–508, 2007) reported that 20 μl of 0.002% TMT elicits an aversive behavior, suggesting that this assay is more sensitive. Indeed, we could see the aversive behavior in Figure 4 using 5 μl of 0.1% TMT in the odor investigation test. Thus, we think that the lack of the ability to observe aversive responses to TMT in the two-odor preference assay and the success in observing the aversion in the odor investigation test are reasonable, considering the results in previous reports.

Figure R4

4) The results of the home cage aversion assay are interesting, and do reveal an important switch in valence at higher concentrations of the Z5 ligand. However, the model put forth in Figure 5, which is a major focus of the manuscript, remains entirely speculative at this time. Are there indeed two additional receptors that can account for this switch? Five? Without any additional evidence the model lacks compelling support.

RESPONSE:

Our data show that Z5-14:OH activates ~6 glomeruli (Figure 2A), suggesting ~3 ORs for Z5-14:OH including Olfr288. But we completely agree that the model remains speculative at this time, unless we could identify the complete set of Z5-14:OH receptors and assess the valence for each receptor. Unfortunately, our trial has so far failed to identify the additional receptors. Therefore, we toned down the conclusion such that we change “whereas the least sensitive (and still-unknown) OR, OlfrY, sends the signal for aversion.” to “whereas the least sensitive (and still-unknown) ORs, a few OlfrY(s), sends the signal for aversion.” (page 13, line 22-23). We also change “three ORs” to “~3 ORs”, and “OlfrX”, “OlfrY” to “OlfrX(s)”, “OlfrY(s)”.

Minor points:

Results, p. 8. Second to last paragraph (related to Figure 2A): no statistical comparison is listed in text or figure/legend. Given the low N and apparent variance in the measurements, this data is not compelling

RESPONSE:

We are sorry for the unclear description. We did the Mann-Whitney test as a statistical analysis, and the P value between WT and KO to 1 mg Z5-14:OH is $P=0.0635$. Because of low N, there was no significant difference but $P=0.0635$ means the tendency to be significant.

Discussion, p. 11, line 13. “muscone and its two ORs” and “Z5-14:OH and its three ORs” – it would be appropriate to clarify each by saying “strongest ORs”

RESPONSE:

We agree with the comment for muscone. For Z5-14:OH, however, it is possible that the least sensitive (but not one of strongest) OR possesses the negative valence. Therefore, we would like to keep it as it is.

Language/typographical:

Intro, p.4, line 4, “is not practical” – redundant phrase

RESPONSE:

We are sorry for the wrong description. We deleted “is not practical”. (page 4, line 8)

Methods, p. 6, line 2-3. “A stream of air was drawn through the charcoal filter from each odorant.”
Confusing. Please clarify

RESPONSE:

We changed the sentences “A stream of air was drawn through the charcoal filter from each odorant.” to “A stream of air was drawn through a charcoal filter from each odorant compartment into the mouse compartment passing along a Y-shaped tube using a vacuum pump.” (page 6, line 5-7).

Figure 4D (figure text): “attracton” should be “attraction”

RESPONSE:

We are sorry for the typo. We changed “attracton” to “attraction” in Figure 1D.

Reviewer #2 (Remarks to the Author):

Major concerns:

Figure 1:

1) The methods section is lacking clear details on how the nose poke was measured and quantified. Was the data obtained visually, using a photodiode, or by automated detection from images? It would be helpful to know whether the animal was required to place its nose through the port or only investigate the immediate vicinity.

RESPONSE:

We are sorry for the unclear description. When the position of the tip of the nose is in the odorant compartment though the hole, we measured that time as the investigation time. Therefore, the data include both the times in poking its nose and in investigating the immediate vicinity. The data were obtained by using a stopwatch manually. We changed sentences “Each test was performed for 5 min, and the amount of time a mouse spent poking its nose through the vent, designated as the investigation time, was measured.” to “Each test was performed for 5 min, and the amount of time a mouse spent poking the tip of its nose through the vent, designated as the investigation time, was measured by using a stopwatch in a double-blind manner.” in Materials and Methods (page 6, line 7-9).

2) Was air blown through each chamber prior to odor being added? Furthermore, was the flow rate from each odor chamber measured and equilibrated? This is especially important if the mouse was required to place his nose through the port as discussed in the previous point.

RESPONSE:

We are sorry for the unclear description. Air was blown through each chamber prior to odor being added, so mice were familiar with airflow before odor comes out. We used a Y-shaped tube to introduce odorants or air control into each compartment, and thus, the airflows in two compartments are always the same. We changed the sentences “A stream of air was drawn through the charcoal filter from each odorant.” to “A stream of air was drawn through a charcoal filter from each odorant compartment into the mouse compartment passing along a Y-shaped tube using a vacuum pump.” (page 6, line 5-7)

Figure 2:

1) Further characterization of the Olf288-ko mouse is necessary. This seems to be a previously unpublished mouse line and there is no supporting evidence to demonstrate that olfr-288 is absent

from the olfactory epithelium, as well as the in the olfactory bulb itself. An in-situ image would be helpful.

RESPONSE:

We are sorry for not including the data. We showed the construction of Olfr288-KO mice (Supplementary Figure 1A, Figure R5 left). No Olfr288 expression was observed by RT-PCR and in situ hybridization in the olfactory epithelium (Supplementary Figure 1B-D, Figure R5 right, Figure R6 below).

Figure R5

Figure R6

2) “freely behaving mice were stimulated through a glass capillary with each odorant for 1 min after

they found the odor source.” - this experimental description lacks key details. It is unclear to this reader how the mice were stimulated with the odor. Self-administration vs forced administration of the odor may result in differences in cFos expression.

RESPONSE:

We are sorry for the unclear description. Mice were stimulated with the odor by using similar way with “Odor-finding test”. Therefore, the way is “Self-administrated”. We changed the sentences “freely behaving mice were stimulated through a glass capillary with each odorant for 1 min after they found the odor source.” to “the tip of a glass capillary with each odorant was set carefully in the home cage. The glass capillary was kept in the home cage for 1 min after freely behaving mice found and began sniffing the odorant. Then the glass capillary with odorant was removed from the home cage carefully.” (page 6, line 14-17).

3) Some example images of cFos staining are necessary. Further, how activated glomeruli were detected and quantified is missing from the methods. There is no discussion of how active glomeruli were detected over those that are activated by background odors in the home cage. Some measurements of the relative positions of the active glomeruli would certainly be helpful to provide some assurance the same ORs are activated in different bulbs/animals.

RESPONSE:

We now show the raw image data of c-Fos staining in Supplementary Figure 2 (Figure R1 below). The arrowheads indicate the activated glomeruli. Regarding the quantification method, we added sentences “Number of activated glomeruli that expressed c-Fos in juxtglomerular cells were counted from one side of the OB from each mouse. No c-Fos induction was observed in the OB of mice with no odor as a control. ” (page 6, line 26-28). The relative positions of the active glomeruli in the olfactory bulb were shown in Supplementary Figure 3 (Figure R2 below). We collected 14 μ m coronal cryosections of the OB, and we plotted the positions of c-Fos positive glomeruli every 5 sections starting from the tip of the anterior OB. Our data suggest that the same ORs are likely activated in different bulbs/animals.

Figure R1

Figure R2

4) Was the number of activated glomeruli counted in each olfactory bulb, or does this number consider the total across both bulbs? If the value is from a single olfactory bulb, was more than one bulb from each mouse considered in the analysis?

RESPONSE:

We are sorry for the unclear description. We counted the number of activate glomeruli in one side of the OB from each mouse. Therefore we added sentences, “Number of activated glomeruli that expressed c-Fos in juxtglomerular cells were counted from one side of the OB from each mouse. No c-Fos induction was observed in the OB of mice with no odor as a control. ” (page 6, line 26-28)

5) Perhaps a more sensitive analysis to consider for the odor-investigation (also the choice task) assay is the latency to find the odor source. For both the WT and KO mice the ability to find the odor source saturates around 1 ng. By measuring the latency to find the odor some deficits may be observed in KO mice. The methods state that this was how the experiment was performed and analyzed, but the data displayed in Figure 2B only present the binary data of success or failure to locate the odor source.

RESPONSE:

We agree that a more sensitive analysis is to measure the latency to find the odor source. We

measured the latency to find the odor source and made a figure (Figure 2B and C bottom, Figure R7 below). The same conclusion was drawn.

Figure R7

Figure 3:

1) The results of the eugenol experiment are surprising. Mice seek novelty and the presence of a neutral odor should elicit some investigation.

RESPONSE:

We agree that the results of the eugenol experiment are somewhat surprising. But it is known that mice like to poke their nose into a hole. Therefore one reason why the investigation time for eugenol and water was the same in the two-choice odor-preference test might be due to the fact that the interests in eugenol and a hole were similar for mice.

2) No aversion is detected to TMT; however, by comparing the number of nose pokes to a period of time prior to the odor delivery, a decrease may be observed. The data appear to be available as there was a 15 minute habituation period for each mouse. This comparison could also help resolve whether a 3 ng concentration of Z5 is aversive. By comparing the relative number of nose pokes prior to and following odor delivery, the authors can correct for any intrinsic spatial biases of individual mice.

RESPONSE:

As this reviewer suggested, we counted the number of nose-pokings into each hole during 5 min before the test period (Supplementary Figure 4, Figure R8 below). There was no significant difference in all figures. Again we could not observe the aversive behavior in this Two-choice odor-preference

test. To correct for any intrinsic special biases of individual mice, we calculated the “preference index”. Preference index means the ratio of investigation time of poking a nose into a hole with a targeted sample to the total investigation time into both holes (Supplementary Figure 5, Figure R9 blow). WT mice significantly preferred Z5-14:OH (10 pg or 1 ng)-spiked castrated urine in comparison to Olfr288-KO mice ($P=0.0049$ for 10 pg, $P=0.0031$ for 1 ng; Mann-Whitney test). There was no significant difference but a tendency for WT mice to prefer 10 pg and 1 ng Z5-14:OH in comparison to Olfr288-KO mice.

Figure R8

Figure R9

3) It is unaddressed as to whether there is a possible trigeminal component to the detection of Z5. In other words, at sufficiently high concentration the odor vapor could generate sensation in the nasal cavity and the mucosa around the eyes. In this scenario, the switch from attraction to aversion could be the result of non-olfactory cues contributing to behavioral output.

RESPONSE:

We used 3 ng Z5-14:OH and 426.4 ng eugenol. The amount of eugenol is much higher than that of Z5-14:OH, but mice did not show any aversive behavior to eugenol. Although it might be different from odorant to odorant, we believe 3 ng Z5-14:OH is too small amount to become a non-olfactory stimulant.

Figure 4:

1) Displaying the position of the nose rather than the center of mass would be beneficial. As displayed it appears that the example mouse avoids, rather than investigates the odor source.

RESPONSE:

We agree that to display the position of the nose is beneficial. We traced the position of the nose and made new figures (Figure R10 below and Figure 4B). We could observe that WT mice smelled a petri dish with 10 pg Z5-14:OH more than Olf288-KO mice.

Figure R10

Minor concerns:

1) Introduction line 4: Olfactory neurons should be changed to olfactory sensory neurons or receptor neurons.

RESPONSE:

We changed “olfactory neurons” into “olfactory sensory neurons”. (page 3, line 6)

2) Introduction paragraph 3: Olfactory neural pathway is ambiguous - this should be explained in some detail.

RESPONSE:

We changed the sentences to “Odorants are recognized by multiple ORs in the olfactory sensory neurons in the olfactory epithelium. Then the odor information is sent to the olfactory bulb (OB) and various higher brain areas, leading to behavioral outputs.” (page 3, line 21-23)

3) Introduction last paragraph, sentence 2: “is not practical” can be removed from the end of the phrase.

RESPONSE:

We removed “is not practical” from the end of the phrase. (page 4, line 8)

4) Methods, Two-choice odor-preference test, sentence 1: Acryl should be changed to acrylic

RESPONSE:

We changed “acryl” into “acrylic”. (page 5, line 32)

5) For each bar graph, showing the distribution of individual points is preferred over simple bars with error.

RESPONSE:

We agree with the comment. We made new figures with individual points (Figure 1 and 3, Figure R11, R12 below).

Figure R11

Figure R12

Reviewer #3 (Remarks to the Author):

1. Throughout the paper, the behavioral assays performed are not particularly sophisticated or sensitive, and seem to be quite variable. This makes the results less convincing and I have doubts about the reproducibility of the findings. For example, in Figure 1, the authors use a two-choice assay to compare the amount of time in a port with control air, propylene glycol, to the time in a port with muscone. The amount of time they spend in the control port ranges from about 11-24 sec, whereas the difference between the two ports is a maximum of about 10 sec, which is within the range of variation. Further, the “significant” result in 1D is so unremarkable that it would go unnoticed if not for the asterisk. What are the standard deviations in these measurements? To their credit, the experiments appear to be performed in a double-blind manner.

RESPONSE:

It turns out that the basal activities are very different from mouse to mouse. To be fair, in the revised manuscript, individual points were given in all figures. Also we put the standard deviation bars in Figure 1D (Figure R13 below). When we focus on one mouse data, the investigation time of muscone was longer than PG in almost all mice. Therefore we think “significant” is correct. We analyzed data in a double-blind manner. We added the words, “in a double-blind manner” (page 6, line 1)

Also we would like to point out that one of the reasons why the significance in WT mice exposed to 45 ng (Figure 1D WT) is smaller than those exposed to 450 μ g (Figure 1C WT) may be that only MOR215-1, but not MOR214-3, is stimulated with 45 ng. Of course, we cannot conclude that the attractive behavior is additive of MOR215-1 and MOR214-3, but this data is suggestive and reasonable.

Figure R13

2. The authors infer the role of untested OSN types in attraction and avoidance behavior, but the conclusions are too strong in the absence of direct evidence. For example, the authors conclude that the attraction to high levels of muscone is due to the activation of MOR214-3 without directly testing

this. The inference is made based on past characterization of musk odor receptors in vitro with heterologous expression. However, it remains possible that another OSN type contributes to the behavior; the past citation is not sufficient to conclude that no other OSNs are activated by the odor. In the case of the elusive Olf_rX & Olf_rY, the authors admittedly don't know the identify of the odorant receptors. Although I agree that the behavioral data is consistent with a model implicating at least two other OSN types, this is speculative and the conclusions should be tempered.

RESPONSE:

We agree that without MOR214-3 knockout/manipulation on top of the MOR215-1 knockout, we cannot conclude that MOR214-3 provides sensitivity/positive valence at higher muscone concentrations. We are sorry that our conclusion was an overstatement. It, however, takes a few years to generate and analyze MOR215-1/MOR214-3 double knockout mice, and even if we did that, it is still possible that the preference behavior does not disappear because it is possible that there is an unidentified muscone OR involved in the attractive behavior. Nonetheless, it is fair to say that regardless of the phenotype of the double knock-out mice, our conclusion that in addition to MOR215-1, there is other OR(s) involved in attraction to muscone is the same. Therefore, we toned down the sentence in Results “Together, these results suggest that both MOR215-1 and MOR214-3 are involved in attraction to muscone (Figure 1C and D, lower panels).” to “Together, these results suggest that MOR215-1 and other receptor(s), likely MOR214-3, are involved in attraction to muscone (Figure 1C and D, lower panels)” (page 9, line 20-21). Further, we clearly indicate that the schematic diagrams in Figure 1C and D, lower panels, are just a model by saying “A possible model” in the legend. We also toned down the sentences in Discussion; “Both MOR215-1 and MOR214-3 are likely to be individually involved in the attractive behavior.” to “MOR215-1 is likely to be involved in the attractive behavior, and MOR214-3 is an additional candidate OR that possesses the positive valence in MOR215-1 KO mice” (page 13, line 16-18). The sentences in Abstract was also toned down; “Muscone is recognized by two ORs, MOR215-1 and MOR214-3, and attracts male mice. Odor preference tests using MOR215-1 knockout mice revealed that both MOR215-1 and MOR214-3 are involved in attraction to muscone.” to “Muscone is recognized by a few ORs including MOR215-1 and MOR214-3, and attracts male mice. Odor preference tests using MOR215-1 knockout mice revealed that both MOR215-1 and other OR(s), likely MOR214-3, are involved in attraction to muscone.”. Regarding Olf_rX and Olf_rY, we toned down the sentence “whereas the least sensitive (and still-unknown) OR, Olf_rY, sends the signal for aversion.” to “whereas the least sensitive (and still-unknown) ORs, Olf_rY(s), sends the signal for aversion.” (page 13, line 22-23). We changed “Olf_rX” and “Olf_rY” in Figure 5 to “Olf_rX(s)” and “Olf_rY(s)”.

3. Fig 2A: The authors use c-fos to assess the number of activated glomeruli by the odorant in wt and Olf288 KO mice. First, they should show some histology for verification of the c-fos detection and evaluation of glomerular identification. Second, it is peculiar that they do this comparison only at a high concentration of odorant, in which there is no behavioral difference between the groups (Fig 2B). Correlating behavior with glomerular activation should be done at the same concentration. Indeed, a careful c-fos analysis of glomerular activation would strengthen their argument about the number of glomeruli or OSN types involved at different concentrations throughout the paper.

RESPONSE:

We agree with the comment. First, we added the raw data of c-Fos staining in Supplementary Figure 2 (Figure R1 below). The arrowheads indicate the activated glomeruli. The relative positions of the active glomeruli in the olfactory bulb were shown in Supplementary Figure 3 (Figure R2 below). Our data suggest that the same ORs are likely activated in different bulbs/animals.

We did the c-Fos staining at a lower concentration (Figure 2A, Figure R3 below). When mice were stimulated with 0.1 ng Z5:14:OH, the number of activated glomeruli in WT and KO was ~2 and ~0, respectively. We should note that the threshold for the KO mice in the odor-finding behavioral test (0.01 ng, Figure 2B) was lower by an order of magnitude. This threshold difference (0.1 ng in c-Fos and 0.01 ng in behavior) could be explained by the previous observation that the threshold of sensitivity in the olfactory bulb imaging is higher than that in the odor-finding behavioral test (Oka et al., *Neuron*, 52:857–869, 2006; Sato-Akuhara et al., *J Neurosci*, 36:4482–4491, 2016). Nonetheless, we can conclude that the Olf288-KO mice are selectively deficient in perceiving low concentrations. We added these data in Figure 2A, and Supplementary Figure 2 and 3.

Figure R1

Figure R2

Figure R3

4. Fig 3 B-E: The statistical analysis is inadequate to support the conclusion that there is a difference between wt and Olf288 KO mice. The authors employ a paired T-test to compare investigation time for each group of mice exposed to two odors, which is appropriate to say that a given group prefers one odor over the other. However, there is no statistical comparison between

groups. The authors conclude that Olf288 is required for attraction to the odorant, but this conclusion must be supported by a statistical analysis comparing the two groups. This seems especially important given that the Olf288 KO mice investigate the odor about as much as the wt mice.

RESPONSE:

We agree with the comment. We calculated the “preference index”. Preference index means the ratio of investigation time of poking a nose into a hole with a targeted sample to that of the total investigation time into both holes (Figure R9 blow). WT mice significantly preferred Z5-14:OH (10 pg or 1 ng)-spiked castrated urine in comparison to Olf288-KO mice (P=0.0049 for 10 pg, P=0.0031 for 1 ng; Mann-Whitney test). There was no significant difference but a tendency for WT mice to prefer 10 pg and 1 ng Z5-14:OH in comparison to Olf288-KO mice.

Figure R9

5. Fig1, 3,& 4: The axis labels, “Investigation time (sec/3 min),” are confusing. Are the data presented as a fraction of total time or as total seconds? The legends state total time, which is in contrast to the axis label.

RESPONSE:

We are sorry for the wrong description. We performed the two-choice odor-preference test for 5 min (Figures 1 and 3) and odor investigation assay for 3 min (Figure 4), and we showed the data as total seconds. We corrected the labels in Figures 1 and 3.

6. Fig 4C: The bottom axis has the label, “milli Q.” What is this? I presumed it was water, but this should be clarified in the figure or legend. Additionally, the dashed line should be defined in the legend.

RESPONSE:

We are sorry for the unclear description. “Milli Q” means water, so we corrected the label in Figure 4C. And we added the sentence “The dashed line in Fig 4C means the average time of water in WT mice” in Materials and Methods. (page7, line 18)

7. The MOR215-1 KO mouse provided by Dr. Sakano is cited, but the cited paper does not have this KO. The paper does have a mouse in which MOR215-1 is replaced by MOR103-1. Is this the mouse used in the current manuscript? Although, technically a deletion, this is more complicated and should be clarified in the manuscript.

RESPONSE:

We are sorry for the unclear description. We used a mouse in which MOR215-1 is replaced by MOR103-1. We added the sentence “The coding sequence of *MOR215-1* was changed to that of *MOR103-1* using recombinant PCR in the *MOR103-1*-> *MOR215-1* IRES *tau-EGFP* mouse (Acc.NO. CDB0545K; <http://www.cdb.riken.jp/arg/mutant%20mice%20list.html>)” in Materials and Methods. (page 5, line 14-16) This mouse is not a typically MOR215-1-deletion mouse. However, when we checked the sensitivity to muscone in these mice by the odor investigation assay before (Sato-Akuhara et al., J Neurosci, 36:4482–4491, 2016), the sensitivity to muscone was reduced. Therefore, we used this mouse as a MOR215-1-KO mouse in this paper.

8. There is an inconsistency between the methods and the figures for the two-choice odor-preference test. Methods state that the test was performed for 5 min whereas the figures state 3 min. This should be clarified.

RESPONSE:

We are sorry for the wrong description. We performed two-choice odor-preference test for 5 min (Figures 1 and 3). We corrected the labels in Figures 1 and 3.

9. In the results section and Fig 5B, the authors state that the Olfr288 KO mice cannot detect odor at 10 pg or less. However, the assay employed cannot distinguish between cannot detect and not interested in the odor. It would be more appropriate to state that mice do not respond, than do not detect.

RESPONSE:

We agree with the comment. Olfr288-KO mice could not detect 10 pg Z5-14:OH in the odor finding test in Figure 2B, however no one knows whether Olfr288-KO mice could not detect 10 pg Z5-14:OH in the two-choice odor-preference test in Figure 3 and the odor investigation assay in Figure 4C. We changed the word “no detection” to “no response” in Figure 5B.

Reviewers' Comments:

Reviewer #1:

Remarks to the Author:

The revised manuscript by Horio, et al. is improved on the original manuscript, which had lacked many critical details. The behavioral approaches used, though each has noteworthy caveats, are presented fairly and the improved communication in the manuscript has had a positive impact. That said, many weaknesses of the original submission remain, limiting confidence the conclusions and the overall impact of the work.

A major concern raised in the initial submission was the conclusion that MOR214-3 is associated with an attraction to muscone at higher concentrations is solely based on the observation that attraction is intact at higher muscone concentrations when MOR215-1 is absent. Because the authors declined to investigate MOR214-3 and MOR215-1 double knockout animals (citing the amount of time and effort required) this conclusion is left in doubt, reducing the overall impact of the findings.

A major concern with the original submission was the lack of proper presentation of the olfactory bulb histological analysis, and the authors responded by adding raw and summary data (Supp. Figs. 2-3) and statistical analysis (Fig. 2A legend). Because of the low number of samples, the statistics do not reach the typical $p < 0.05$ threshold. Although the over-reliance on this arbitrary cutoff is problematic, in this case the lack of statistical power causes considerable concern about whether these results can be used as the basis for a strong conclusion (e.g. associating the approximate number of glomeruli with an approximate number of ORs participating in an odorant response).

In the rebuttal, the authors' explanation for the lack of aversion to TMT in the choice assay is hard to follow. At best, the explanations are speculative. Even with these explanations, these data raise more questions than they provide answers.

In the end, the manuscript provide plausible models for integration of information from receptors for these ligands, but the new experimental support for these models is still weak.

Reviewer #2:

Remarks to the Author:

The authors have answered all my queries, and clarified some issues. I don't have additional comments.

Reviewer #3:

Remarks to the Author:

The revised manuscript is much improved. The authors have satisfied all of my previous concerns and the manuscript is now suitable for publication.

I have one minor comment to improve the manuscript. In my previous concern (#1), I questioned the robustness of the behavior. The authors wrote in response: "When we focus on one mouse data, the investigation time of muscone was longer than PG in almost all mice." I think this is an important statement that mitigates concern over variability in the data, and I recommend including such a statement in the results section for Figure 1.

Reviewer #1 (Remarks to the Author):

The revised manuscript by Horio, et al. is improved on the original manuscript, which had lacked many critical details. The behavioral approaches used, though each has noteworthy caveats, are presented fairly and the improved communication in the manuscript has had a positive impact. That said, many weaknesses of the original submission remain, limiting confidence the conclusions and the overall impact of the work.

1) A major concern raised in the initial submission was the conclusion that MOR214-3 is associated with an attraction to muscone at higher concentrations is solely based on the observation that attraction is intact at higher muscone concentrations when MOR215-1 is absent. Because the authors declined to investigate MOR214-3 and MOR215-1 double knockout animals (citing the amount of time and effort required) this conclusion is left in doubt, reducing the overall impact of the findings.

RESPONSE:

We agree that we cannot conclude the involvement of MOR214-3 without MOR215-1/214-3 knockout mice data, but also an outcome phenotype may not be conclusive due to involvement of other receptor(s). Thus, in the final revision, we decided to further tone down sentences, “Odor preference tests using MOR215-1 knockout mice revealed both MOR215-1 and other OR(s), likely MOR214-3, are involved in attraction.” to “Odor preference tests using MOR215-1 knockout mice revealed MOR215-1 and other OR(s), possibly including MOR214-3, are involved in attraction.” (page 2 , line 5-6), “Together, these results suggest that MOR215-1 and the other receptor(s), likely MOR214-3, are involved in attraction to muscone (Figure 1C and D, lower panels)” to “Together, these results suggest that MOR215-1 and the other receptor(s), possibly including MOR214-3, are involved in attraction to muscone (Figure 1C and D, lower panels).” (page 5, line 21-22) , and “A possible model showing the involvement of two muscone ORs in attraction to 450 μ g muscone.” to “A possible model showing the involvement of at least two muscone ORs in attraction to 450 μ g muscone.” (page 19, line 13-14)

2) A major concern with the original submission was the lack of proper presentation of the olfactory bulb histological analysis, and the authors responded by adding raw and summary data (Supp. Figs. 2-3) and statistical analysis (Fig. 2A legend). Because of the low number of samples, the statistics do not reach the typical $p < 0.05$ threshold. Although the over-reliance on this arbitrary cutoff is problematic, in this case the lack of statistical power causes considerable concern about whether these results can be used as the basis for a strong conclusion (e.g. associating the approximate number of glomeruli with an approximate number of ORs participating in an odorant response).

RESPONSE:

We agree that because of the low number of samples, the statistical power is small and we have a concern about whether these results can be used as the basis for a strong conclusion. We calculated the effect size (Cohen's *d*) of 1 mg and 0.1 ng Z5-14:OH (N=5 in WT and KO each) at first, and the effect size of 1 mg Z5-14:OH and 0.1 ng Z5-14:OH was 1.89 and 1.64, respectively. Then we calculated the statistical power with these effect sizes, and the statistical power of 1 mg and 0.1 ng Z5-14:OH was 0.72 and 0.47, respectively. More than 0.8 of the statistical power is recommended to make a conclusion, and as this reviewer mentioned, our statistical power is lower than recommended. We calculated the recommended sample size with these effect sizes to get 0.8 of the statistical power, and we found the recommended sample size was 6 for 1 mg Z5-14:OH and 8 for 0.1 ng Z5-14:OH. Thus, we decided to add additional samples.

It turns out that the c-Fos antibody that we used and worked well is no more available. We then tried several new lots of c-Fos antibodies, but we could not find an antibody that worked well for this purpose. Therefore, we performed c-Fos in situ hybridization to get the number of c-Fos positive glomeruli. In the new figure 2A and R1, we plotted the data with a triangle symbol for the immunohistochemistry result and with a cross symbol for the in situ hybridization result. After adding N, the statistical power of 1 mg and 0.1 ng Z5-14:OH became 0.89 and 0.82, respectively. Significant differences were observed between WT and Olfr288-KO in both 1 mg and 0.1 ng Z5-14:OH (P=0.0281 and P=0.0130, respectively). Although theoretically, there should be no difference in numbers of c-fos positive glomeruli obtained by immunohistochemistry or in situ hybridization, it is better to be argued with the same method, but unfortunately, we have had a breeding problem and we have not gotten enough number of Olfr288-KO mice and will take another half year or so to add enough numbers of in situ experiments. The quantity of the difference was about 2 in both of 1 mg and 0.1 ng Z5-14:OH groups, consistent with the knowledge that axons of sensory neurons expressing the same OR converge onto typically two glomeruli in one side of the OB. Thus, associating the approximate number of glomeruli with an approximate number of ORs participating in an odorant response appears to be reasonable, and it can be concluded that Olfr288 is the most sensitive OR for Z5-14:OH among ~3 ORs in mice.

Figure R1

3) In the rebuttal, the authors' explanation for the lack of aversion to TMT in the choice assay is hard to follow. At best, the explanations are speculative. Even with these explanations, these data raise more questions than they provide answers.

RESPONSE:

We are very sorry for our poor description. We did not write the amount of TMT in Figure 3, and in this revision, we added the description in Figure 3. Also we compared the amount of TMT that was used in our experiments and previous reports (Table R1 below). In a previous report (Saito et al., Nat Commun., 8:16011, 2017), it was said that 10 μ l of 100% TMT caused the immobility in WT mice, but the immobility with 1 μ l of 100% TMT was much smaller than 10 μ l of 100% TMT. In our two-choice odor-preference test, we could not see the immobility because our amount of TMT (5 μ l of 0.1% TMT, which was the same as 5 nl of 100% TMT) was 200-2000 times lower than those in the Saito et al paper. We cannot exactly compare their study and ours, but it is possible that the amount of TMT might not be enough to see the immobility behavior in our two-choice odor-preference test.

In contrast, our odor investigation test in Figure 4 is the similar method as utilized in Kobayakawa et al. paper (Nature, 450:503–508, 2007). They used 20 μ l of 0.002% TMT (corresponding to 0.4 nl of 100% TMT) that elicited an aversive behavior. We cannot exactly compare the immobility test in Saito et al. and odor investigation test in Kobayakawa et al., but the amount of TMT in Saito et al. was 2500-25000 times larger than that in Kobayakawa et al., suggesting that the odor investigation test is more sensitive than the immobility test. We used about 10 times larger amount of TMT (5 μ l of 0.1% TMT that means 5 nl of 100% TMT) than that used in Kobayakawa et al., and thus, could see the aversive behavior.

	Description of TMT concentration and volume in the paper	TMT volume if TMT is 100%	Phenotype
our two-choice odor-preference test (Figure 3A)	5 μ l of 0.1% TMT	5 nl of 100% TMT	No immobility, no aversive behavior
Saito et al. (Nature Communi. 8:16011, 2017)	100 μ l of 10% TMT	10 μ l of 100% TMT	Large immobility (average immobility time 2.5 min per 5min)
	100 μ l of 1% TMT	1 μ l of 100% TMT	Small immobility (average immobility time 0.5 min per 5min)
our odor investigation test (Figure 4A)	5 μ l of 0.1% TMT	5 nl of 100% TMT	aversive behavior
Kobayakawa et al., (Nature, 450:503–508, 2007)	20 μ l of 0.002% TMT	0.4 nl of 100% TMT	aversive behavior

Table R1

4) In the end, the manuscript provide plausible models for integration of information from receptors for these ligands, but the new experimental support for these models is still weak.

RESPONSE:

We agree that the new experimental support for these models is still weak because we have not generated MOR215-1/MOR214-3 double knockout mice and we have not obtained additional Z5-14:OH receptors. Nonetheless, we have circumstantial evidences to say the following conclusion (in Abstract) “The final output behavior with multiple ORs stimulation is determined by summation (addition or competition) of valences coded by activated ORs.” We believe this concept will be further examined for various receptors among hundreds to thousand members of ORs in the future.

Reviewer #2 (Remarks to the Author):

The authors have answered all my queries, and clarified some issues. I don't have additional comments.

RESPONSE:

We appreciate this comment.

Reviewer #3 (Remarks to the Author):

The revised manuscript is much improved. The authors have satisfied all of my previous concerns and the manuscript is now suitable for publication.

I have one minor comment to improve the manuscript. In my previous concern (#1), I questioned the robustness of the behavior. The authors wrote in response: "When we focus on each mouse data, the investigation time of muscone was longer than PG in almost all mice." I think this is an important statement that mitigates concern over variability in the data, and I recommend including such a statement in the results section for Figure 1.

RESPONSE:

We agree with this comment. We added the sentence, “The basal activities are very different from mouse to mouse, however when we focus on one mouse data, the investigation time of muscone was longer than PG in almost all WT male mice.” (page 5, line 13)